# Epidemiological Characteristics of Pediatric Patients with Intestinal Failure in Spain: Data from the REPAFI Registry

**DOI:** 10.3390/nu17233768

**Published:** 2025-11-30

**Authors:** Marta Germán-Díaz, Rocío González-Sacristán, Vanessa Cabello, Javier Blasco-Alonso, Alejandro Rodríguez, Mariela de los Santos, José Vicente Arcos-Machancoses, Mercedes Murray-Hurtado, Ruth García-Romero, Rafael Galera-Martínez, Cristina Martín-Arriscado, Susana Redecillas-Ferreiro, José Manuel Moreno-Villares, Esther Ramos-Boluda

**Affiliations:** 1Pediatric Gastroenterology, Hepatology and Nutrition Department, Hospital Universitario 12 de Octubre, 28041 Madrid, Spain; 2Intestinal Rehabilitation Unit, Hospital Universitario La Paz, 28046 Madrid, Spainerboluda@salud.madrid.org (E.R.-B.); 3Pediatric Gastroenterology and Nutrition Support Unit, Hospital Universitario Vall D’Hebron, 08035 Barcelona, Spain; 4Pediatric Gastroenterology and Nutrition Department, Hospital Regional Universitario de Málaga, 29010 Málaga, Spain; javierblascoalonso@yahoo.es; 5Pediatric Gastroenterology Department, Hospital Universitario Virgen del Rocío, 41013 Sevilla, Spain; 6Pediatric Gastroenterology, Hepatology and Nutrition Department, Hospital Universitario Sant Joan de Deu, 08950 Barcelona, Spain; 7Department of Pediatric Gastroenterology, Hepatology and Nutrition, Hospital Clínic Universitari, 46014 Valencia, Spain; 8Pediatric Nutrition and Inborn Errors of Metabolism Section, Complejo Hospitalario Universitario de Canarias, 38320 San Cristóbal de La Laguna, Spain; 9Pediatric Gastroenterology and Nutrition Department, Hospital Universitario Miguel Servet, 50008 Zaragoza, Spain; ruthgarciaromero@yahoo.es; 10Pediatric Gastroenterology and Nutrition Department, Hospital Universitario Torrecárdenas, 04009 Almería, Spain; 11Research Institute, Hospital Universitario 12 de Octubre, 28041 Madrid, Spain; 12Pediatrics Departement, Clínica Universitaria de Navarra, 28027 Madrid, Spain; jmorenov@unav.es

**Keywords:** intestinal failure, home parenteral nutrition, children, epidemiology, registry

## Abstract

Background: The paucity of data on the epidemiology of chronic intestinal failure (CIF) in pediatric patients is a matter of particular concern. The objective of this article is to provide a comprehensive description of the epidemiology of CIF in Spain, encompassing its incidence, geographical distribution, underlying causes, and demographic and clinical characteristics. These findings are based on data collected from the multicentre REPAFI registry. Methods: This is a national, multicentre, ambispective cohort study including patients who initiated home parenteral nutrition (HPN) between January 2015 and January 2025. The data collected encompassed various demographic details, underlying diagnoses, the type of HPN utilized, and the nutritional status of the subjects at the commencement of HPN treatment. Results: The study included 163 patients (55.2% male) from 10 hospitals. The principal cause of CIF was short bowel syndrome (SBS) in 77.3% of cases, followed by severe motility disorders (12.9%), congenital enteropathies (CE) (5.5%), and other causes (4.3%). Among patients diagnosed with SBS, necrotizing enterocolitis was identified as the most prevalent underlying cause (32.5%). The most prevalent anatomical configuration was identified as type 2 (jejuno-colic anastomosis). A significant proportion, amounting to 62.7%, exhibited a lack of an ileocecal valve (ICV), while 23% demonstrated a residual bowel length (RBL) of less than 15 centimetres. The median RBL was 35 cm (IQR: 15.7–52.5). Patients diagnosed with SBS exhibited a lower gestational age and birthweight compared with the other groups (*p* < 0.05). Patients diagnosed with SBS and CE exhibited a lower mean age at the onset of HPN (*p* < 0.05). Furthermore, patients with CE exhibited the lowest weight-for-age Z-score at the initiation of HPN (*p* < 0.05). Conclusions: The present study provides the first epidemiological data on the state of pediatric CIF in Spain. The most prevalent cause of CIF was SBS, with a younger age at the initiation of HPN in comparison to other published studies. Patients with CE exhibited the most severe degree of malnutrition at the initiation of HPN.

## 1. Introduction

Intestinal failure (IF) is defined as a situation in which a critical reduction in functional gut mass to below the minimum needed to absorb liquids and nutrients prevents adequate growth in children and weight maintenance in adults [1]. IF can be classified into three groups based on its pathophysiology: short bowel syndrome (SBS; the most common cause of IF in infancy), motility disorders, and congenital enteropathies [1,2,3,4]. In cases of chronic or irreversible IF, home parenteral nutrition (HPN) provides an alternative to prolonged hospitalization and is currently regarded as the optimal treatment option for these patients. This assertion is supported by evidence that demonstrates the efficacy of HPN in enhancing the quality of life for both patients and their families [5,6]. In recent years, considerable progress has been witnessed in the multidisciplinary management of these patients, including the development of novel surgical techniques (including intestinal transplantation), advancements in the composition of parenteral nutrition solutions, and improvements in the care of central venous catheters. Additionally, the new drugs have emerged that enhance survival rates and quality of life for children afflicted with this condition [2,3,7,8]. Despite these advances, the chronic character of this pathology, as well as the need for the prolonged administration of parenteral nutrition through a central venous catheter, contribute to the onset of complications such as catheter-related infections and intestinal failure-associated liver disease which substantially impact the prognosis of these patients.

The low prevalence of this condition poses a significant challenge in the development of evidence-based protocols or clinical practice guidelines, which are essential for standardizing the management of these patients. Indeed, there is a paucity of data on the epidemiology of this condition in pediatric patients, although several publications have reported data from national registries in our region [9,10,11,12,13,14,15]. The prevalence of pediatric IF ranges from 9.56 cases per million inhabitants in Germany to 27 per million inhabitants in Portugal. A significant proportion of these longitudinal studies have drawn attention to the increase in the prevalence of pediatric IF in recent years, attributing this phenomenon to a variety of factors. These include enhanced survival rates among patients with chronic IF (CIF) or SBS, a decline in mortality among patients on HPN, advancements in the prevention and treatment of intestinal failure-related liver disease, and the establishment of multidisciplinary treatment teams that oversee the management of these patients [13,15].

In Spain, the Home and Ambulatory Artificial Nutrition (NADYA) Group of the Spanish Society of Clinical Nutrition and Metabolism (SENPE) is responsible for maintaining an HPN registry, which encompasses both adult and pediatric patients. Their most recent report, published in 2019, documented 31 cases of pediatric patients [16]. However, it is reasonable to hypothesize that the true incidence is considerably higher than the reported figure, given that only three pediatric centres participated in the registry. In order to address this need, the Spanish Society of Pediatric Gastroenterology, Hepatology and Nutrition (SEGHNP) established the national pediatric registry for chronic intestinal failure (REPAFI). The objective of this initiative was to enhance the comprehension of this condition despite its infrequent prevalence. From that initial standpoint, concerted management strategies may be formulated to establish standardized follow-up protocols and ensure consistency in patient care and outcome assessment.

The objective of the present article is to provide a comprehensive description of the epidemiology of CIF in our country. The description will encompass the following aspects: incidence, geographical distribution, underlying causes, and demographic and clinical characteristics. These aspects will be supported by data collected from the REPAFI registry.

## 2. Materials and Methods

This is a national, multicentre, ambispective cohort study including patients who initiated HPN between January 2015 and January 2025.

### 2.1. Subjects

Children from 0 to 18 years old who met the diagnostic criteria for CIF, had received parenteral nutrition for at least 90 days [7], and begun HPN as of January 2015. Parenteral nutrition is understood as a mixture of amino acids, glucose, lipids, electrolytes, vitamins, and trace elements.

### 2.2. Data Collection Methods

The data for this study were collected and manipulated using the online data collection tool REDCap, sponsored by the Spanish Society of Pediatric Gastroenterology, Hepatology and Nutrition (SEGHNP) and with technical support from AEGREDCap (The Spanish Association of Gastroenterology) [17,18]. REDCap (Research Electronic Data Capture) is a secure web application designed for research data management.

Data were collected regarding demographics, the underlying disease behind CIF, the type of HPN used, and the nutritional situation at the beginning of HPN.

The data concerning patients who initiated HPN between January 2015 and January 2025 were collected retrospectively until January 2020. By contrast, data pertaining to patients who initiated HPN between January 2020 and January 2025 were collected prospectively.

### 2.3. Statistical Analysis

The REDCap database was used to collate the variables, which were then analyzed using SPSS Statistics 25 and STATA 18 software. The performance of descriptive statistics was undertaken with absolute frequencies and percentages for qualitative data being utilized. The median and upper and lower quartiles were utilized for the analysis of quantitative data. The comparison between the different groups of causes of CIF was carried out using the non-parametric Kruskal–Wallis test, followed by a post hoc correction to perform two-by-two comparisons between groups.

### 2.4. Ethical Considerations

In order to be included in the registry, parents or legal guardians were required to provide their written informed consent. The project was approved by the Clinical Research Ethics Committee of 12 de Octubre University Hospital (Nº CEIm: 19/542).

## 3. Results

The total number of patients added to the registry during the study period was 163 (55.2% male), drawn from 10 hospitals. The distribution of patients by referring hospital is demonstrated in Figure 1. The principal cause of CIF was SBS, identified in 77.3% of cases, followed by severe motility disorders (12.9%), congenital enteropathies (CE) (5.5%), and other causes (4.3%) (Table 1). The distribution of the causes of CIF was similar across all centers, with minor variations. In all cases, SBS was by far the most prevalent cause. Among patients with SBS, necrotizing enterocolitis was the most prevalent underlying cause (32.5%), followed by intestinal volvulus (19%), intestinal atresia (17.5%), and gastroschisis (13.5%) (Figure 2). The most frequent anatomical type was type 2 (jejuno-colic anastomosis), while 62.7% of the total cohort of patients with SBS exhibited a lack of an ileocecal valve (ICV) and 23% had a residual bowel length (RBL) of less than 15 cm (Table 2). The median RBL was 35 cm (IQR: 15.7–52.5), with a range spanning from 0 to 225 cm. The median age at the start of HPN was 6.10 months (IQR: 4.40–9.50).

Table 3 shows the remaining causes of CIF other than SBS.

The number of patients who started HNP each year varied from 8 in 2016 and 2017 to 23 in 2019 and 2022, with the incidence of CIF varying from 1.47 cases per million inhabitants < 18 in 2015 to 2.85 cases per million inhabitants in 2022, the most recent year for which data from the National Statistics Institute is available regarding the at-risk population. Table 4 presents the characteristics of the patients added to the registry each year.

Furthermore, an attempt was made to ascertain whether there were any discrepancies with regard to gestational age, birth weight, age at the onset of HPN, and the nutritional status of the patient at the start of HPN in accordance with the underlying cause of CIF. The findings of this study demonstrate that patients diagnosed with SBS have a lower gestational age and birthweight compared with the other groups. The age at which HPN is initiated is lower in patients with CIF and CE. Furthermore, patients with CE exhibit the lowest weight-for-age Z-score at the initiation of HPN (see Table 5).

## 4. Discussion

For patients diagnosed with CIF, HPN represents the optimal therapeutic intervention following a variable period of inpatient intestinal rehabilitation [19,20]. This nutritional support method was first established in the late 1960s in the United States and the early 1970s in some European countries. Denmark was the first European country to implement HPN programs, initiating them in 1970. The first case of HPN in France was reported in the early 1970s, while in the United Kingdom, initial cases emerged in the late 1970s, followed by a steady increase in subsequent decades. In other European countries, such as Belgium and Germany, the expansion of HPN occurred primarily during the 1980s and 1990s [21]. Since that time, HPN has gradually gained acceptance as a therapeutic option for patients suffering from prolonged or permanent IF, though with notable variations across different countries. Spain’s implementation of HPN occurred at a later date than in other countries; the first adult patients were discharged in the late 1980s [22], followed by the first pediatric patients in the early 1990s [23]. In Spain, PN is included in the common range of specialized services offered both in inpatient and outpatient (HPN) settings, and is therefore covered by the Public National Health System [24]. However, in contrast to home enteral nutrition (HEN), there are no regulatory guidelines standardizing HPN.

The true incidence and prevalence of HPN in pediatric patients in Spain remains to be elucidated. In contrast to the systems in place in the United Kingdom, Denmark, or France, where patients receiving HPN are funnelled into reference hospitals spread out throughout the country, in Spain, practically any hospital can have patients on HPN. This approach introduces challenges in the collection of data. In order to estimate the necessary resources for the implementation of this therapeutic approach, it is first necessary to ascertain the number of patients requiring HPN. One method of approximating this need is through the utilization of data from national patient registries [25]. This study presents data from Spanish pediatric HPN patients for the first time. The study’s participants were drawn from a cohort of 163 patients from 10 hospitals nationwide who had initiated HPN over a 10-year period. The annual number of patients commencing HPN varied between 8 and 23, with the incidence of CIF ranging from 1.47 cases per million inhabitants under the age of 18 in 2015 to 2.85 cases per million inhabitants in 2022, the most recent year for which data from the National Statistics Institute is available concerning the at-risk population. These data are consistent with those from neighbouring countries [2] and with a general trend noted in recent years of an increasing prevalence of pediatric CIF [2,13,15]. This phenomenon has been attributed to an increase in patients with non-primary digestive diseases, attributable to the enhanced survival rates of patients afflicted with complex chronic conditions. This includes an increased number of patients with neurological damage and associated motility disorders, such as those with cerebral palsy. However, in the present study, the percentage of patients with non-primary digestive diseases was low (4.3% of the total), which is lower than in other studies [2,13]. Furthermore, there were no patients fitting this profile who required HPN. The most prevalent cause of CIF was SBS, with necrotizing enterocolitis being the most common cause of SBS, as evidenced by the extant literature [1,2,3,13,14,15]. The median RBL was found to be 35, while the median age at the initiation of HPN across the entire study was 6.6 months, which is lower than that reported in other studies [13,19,26]. For instance, in the Colomb study, the median age at the initiation of HPN was 1.5 years [19], whereas in Gandullia it was 2.35 years [26]. The most recent report from the French registry demonstrated a decline in the median age at the initiation of HPN, from 11.7 months in 2014 to 8.3 months in 2019 [13]. This finding indicates that hospitals are identifying patients eligible for HPN earlier, thereby reducing the duration of their hospital stays and the risk of potential complications. In the present study, the age at which HPN was initiated remained consistent over the 10-year period, ranging from 5 to 8 months of age. As would be anticipated, patients with SBS and CE exhibited a lower age at the onset of HPN compared with those with SMD and other pathologies. It was hypothesized that patients with SBS would demonstrate a lower gestational age and birth weight compared with the other groups. Finally, it was observed that patients with congenital enteropathies exhibited greater levels of malnutrition at the initiation of HPN in comparison to the other groups, as evidenced by a lower weight Z-score. This provides a quantitative indication of the severity of this particular subgroup of patients suffering from such rare pathologies.

The principal limitation of the present study is that it uses a voluntary registry; as such, it is likely that it does not include all pediatric patients who initiated HPN in our country over the last decade. Nevertheless, we consider our sample to be representative of the target population, as it included the participation of all the major hospitals in Spain that treat patients with this profile As mentioned before, in our country, there are no centres which are officially accredited to treat this profile of patients. It is acknowledged by the Intestinal Failure Research Group of SEGHNP that the management of this patient population in multidisciplinary units specializing in intestinal rehabilitation is of paramount importance. This approach has been shown to be associated with a reduction in morbidity and mortality [27]. Consequently, in 2020, we published two documents. The first constituted consensus guidelines that delineated the criteria referral centres for IF had to adhere to. The second comprised referral criteria for pediatric intestinal rehabilitation programmes. Both documents are available on the society’s website https://www.seghnp.org/grupos-trabajo/fracaso-intestinal (accessed on 24 November 2025). It is noteworthy that there is only one centre in the entire country that performs pediatric intestinal transplants, which also serves as the leading reference hospital for IF, namely the University Hospital La Paz in Madrid. This finding is consistent with the observation that a significant proportion (57.7%) of the patient population included in the present study was affiliated with that particular medical institution. However, it is important to note that there are also other multidisciplinary intestinal rehabilitation units in the country with extensive experience in the management of pediatric patients with CIF.

A further limitation of the present study is that it exclusively included patients who initiated HPN during the study period; data from patients who had already initiated HPN prior to the study period and continued treatment were not collected. Consequently, the collection of data on disease prevalence was not possible, only incidence could be calculated.

## 5. Conclusions

Our study provides epidemiological data on the state of IF in Spain, being the first study to detail the use of HPN in pediatric patients in our country. Our findings are similar to those observed in nearby countries such as Italy or the United Kingdom, with an incidence of CIF around 1.47–2.85 cases per million inhabitants < 18. The most frequent cause of CIF was SBS, with a younger age at the start of HPN compared with other published studies. Patients with congenital enteropathies presented the highest degree of malnutrition at the start of HPN.

## Figures and Tables

**Figure 1 nutrients-17-03768-f001:**
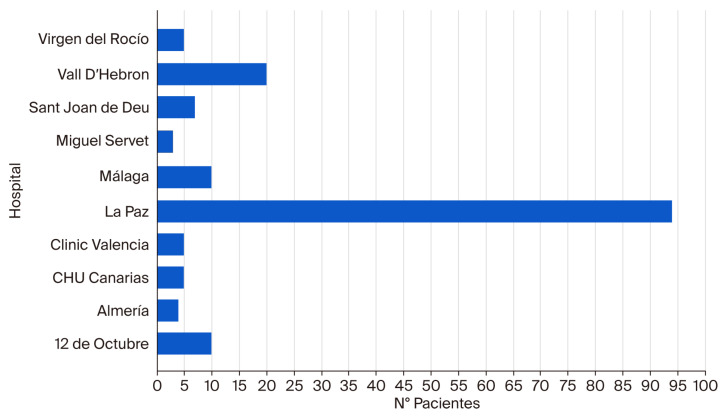
Distribution of patients enrolled in the registry by hospital of origin.

**Figure 2 nutrients-17-03768-f002:**
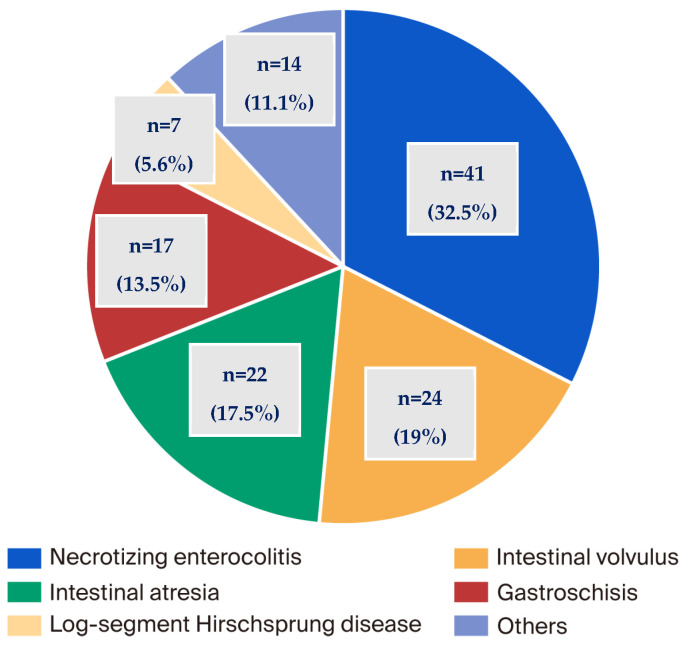
Causes of short bowel syndrome (SBS).

**Table 1 nutrients-17-03768-t001:** Causes of chronic intestinal failure (CIF).

	N° Patients	Percentage
Short Bowel	126	77.3
Congenital Enteropathy	9	5.5
Severe Motility Disorder	21	12.9
Other (Immunodeficiencies, inflammatory bowel disease, cancer, cystic fibrosis…)	7	4.3
Total	163	100.0

**Table 2 nutrients-17-03768-t002:** Anatomical characteristics of patients with short bowel syndrome (SBS).

	N° Patients	Percentage
Anatomical Type		
Type 1 (enterostomy)	32	25.4
Type 2 (jejuno-colic anastomosis)	59	46.8
Type 3 (jejuno-ileal anastomosis with ICV)	34	27
Classification According to RBL		
<15 cm	29	23
15–40 cm	46	36.5
>40 cm	50	39.7
Presence of ICV		
No	79	62.7
Yes	44	34.9

ICV: ileocecal valve. RBL: remaining bowel length.

**Table 3 nutrients-17-03768-t003:** Diagnoses of patients with causes of chronic intestinal failure (CIF) other than short bowel syndrome (SBS).

Diagnosis	N° Patients
**Severe Motility Disorders**	**21**
Chronic intestinal pseudo-obstruction	13
Others	8
**Congenital Enteropathies**	**9**
Microvillus inclusion disease	3
Intestinal epithelial dysplasia	3
Trichohepatoenteric syndrome	2
Others	1
**Other Causes**	**7**
Syndromes	2
Cancer	2
Chylomicron retention disease	1
Short bowel resection with associated motility disorder	1
Sclerosing peritonitis	1

**Table 4 nutrients-17-03768-t004:** Characteristics of patients enrolled in the registry each year.

Year	2015	2016	2017	2018	2019	2020	2021	2022	2023	2024
N° patients	11	8	8	20	23	17	22	23	16	15
Sex (%)										
Male	81.8	50	62.5	40	65.2	47.1	40.9	65.2	43.8	60
Female	18.2	50	37.5	60	34.8	52.9	59.1	34.8	56.3	40
Gestational Age										
Median (weeks) [IQR]	36 [33.0–39.0]	32 [29.5–38.8]	34 [27.5–39.5]	35 [27.8–38.0]	35 [34.0–39.0]	35 [27.0–39.0]	38 [33.8–39.0]	35 [33.0–36.3]	36 [34.0–38.0]	31 [28.0–37.0]
Birth Weight										
Median (kg)	2.5	2.16	2.09	2.01	2.14	2.5	2.82	2.21	2.52	1.67
[IQR]	[2.0–2.8]	[1.2–3.3]	[1.1–3.1]	[0.9–2.8]	[1.7–2.9]	[1.0–3.0]	[1.9–3.1]	[1.6–3.0]	[1.7–3.2]	[1.0–2.5]
Length at Birth										
Median (cm)	45	44	43	44	45	45.5	47	44.5	46	42.5
[IQR]	[41–49]	[35–50]	[35–49]	[35–49]	[40–47]	[35–50]	[45–50]	[42–48]	[44–49]	[35–46]
Cause of CIF (%)										
SBS	72.7	87.5	87.5	75	78.3	82.4	63.6	82.6	75	80
CE	9.1	0	0	0	17.4	5.9	0	0	0	20
SMD	18.2	12.5	12.5	25	4.3	5.9	22.7	8.7	18.8	0
Other	0	0	0	0	0	5.9	13.6	8.7	6.3	0
RBL										
Median (cm)	36	30	26	29	25	48	34	48	55	32
[IQR]	[31.3–47.5]	[8.0–48.0]	[5.0–40.0]	[15.2–42.5]	[11.5–32.5]	[12.0–72.5]	[6.2–51.5]	[28.0–90.0]	[32.5–109.0]	[20.5–50.0]
Age at Start of HPN										
Median (months)	6	8.3	6.3	6.8	7.9	7.2	6.8	6.1	5.3	6.2
[IQR]	[3.3–8.5]	[3.2–17.3]	[4.1–8.1]	[5.2–9.0]	[4.8–14.4]	[5.0–15.6]	[5.5–32.0]	[3.8–27.1]	[3.4–67.4]	[4.0–18.7]
Weight-for-age Z-score at the start of HPN										
Median	−3.6	−3.7	−3.9	−4.3	−3.9	−3.6	−2.8	−2.6	−3.3	−3.9
[IQR]	[−5.0–−2.3]	[−4.9–−2.6]	[−5.6–−2.6]	[−5.4–−1.6]	[−5.3–−2.7]	[−5.4–−2.2]	[−3.9–−1.6]	[−4.1–−1.3]	[−4.6–−1.8]	[−5.3–−3.1]
Length-for-age Z-score at the start of HPN										
Median	−2.6	−1.8	−2.3	−4	−3.7	−3.4	−1.8	−2.3	−2.6	−3.8
[IQR]	[−4.3–−2.0]	[−5.1–−1.5]	[−6.5–−0.7]	[−6.4–−2.2]	[−4.5–−2.0]	[−4.3–−1.9]	[−3.0–−0.3]	[−3.7–−0.7]	[−3.8–−0.4]	[−4.9–−2.7]
Body Mass Index (BMI)-for-age Z-score at the start of HPN										
Median	−2	−3.3	−2.7	−1.8	−2.7	−2	−2.2	−1.6	−2.2	−2.4
[IQR]	[−4.1–−1.4]	[−3.6–−0.5]	[−4.6–−2.3]	[−3.8–−1.1]	[−4.0–−1.4]	[−3.8–−1.2]	[−3.0–−0.8]	[−3.7–−0.5]	[−3.9–−1.2]	[−3.2–−1.2]
Type of PN administered (%)										
Individually prepared by hospital pharmacy	45.5	25	37.5	45	26.1	35.3	36.4	30.4	56.3	60
Individually prepared by catering service	54.5	75	62.5	55	73.9	64.7	63.6	69.6	43.8	40
Standard	0	0	0	0	0	0	0	0	0	0

BMI: body mass index. CE: congenital enteropathy. CIF: chronic intestinal failure. HPN: home parenteral nutrition. IQR: interquartile range. PN: parenteral nutrition. SBS: short bowel syndrome. SMD: severe motility disorder.

**Table 5 nutrients-17-03768-t005:** Comparison between the different causes of chronic intestinal failure (CIF). The comparison between the different groups of causes of CIF was carried out using the non-parametric Kruskal–Wallis test, followed by a post hoc correction to perform two-by-two comparisons between groups. A *p*-value of <0.05 (have been bold) was considered statistically significant.

Characteristics		Cause of CIF					Comparison by Groups (*p*-Value)					
	Total	SBS (1)	CE (2)	SMD (3)	Others (4)	*p*-Value	*p* 1–2	*p* 1–3	*p* 1–4	*p* 2–3	*p* 2–4	*p* 3–4
	N = 163	N = 126	N = 9	N = 21	N = 7							
GA						**<0.01**	**0.02**	**<0.01**	**0.04**	0.94	0.75	0.8
Median (weeks)	35	34	38	38	38							
[IQR]	[31.5–38.0]	[30.0–37.0]	[37.0–39.0]	[37.0–39.0]	[34.0–40.0]							
BW						**0.04**	0.07	**0.03**	0.23	0.86	0.92	0.75
Median (kg)	2.32	2.15	2.66	2.71	2.8							
[IQR]	[1.63–2.95]	[1.43–2.89]	[2.48–2.80]	[2.27–3.10]	[1.75–3.26]							
Age at Start of HPN						0.06	0.28	**0.02**	**0.01**	0.27	0.06	0.61
Median (months)	6.6	6.1	8	25	17							
[IQR]	[4.60–15.00]	[4.40–9.50]	[6.20–14.40]	[5.10–105.80]	[11.90–101.00]							
Weight-for-age Z-score at the start of HPN						**0.04**	0.18	**0.04**	0.19	**0.01**	**0.04**	0.92
Median	−3.4	−3.5	−4.3	−2.5	−2.8							
[IQR]	[−5.00–−2.30]	[−5.10–−2.40]	[−4.90–−3.70]	[−3.70–−1.70]	[−3.20–−0.70]							
Length-for-age Z-score at the start of HPN						0.24	0.21	0.23	0.52	**0.03**	0.09	0.89
Median	−2.85	−2.9	−3.8	−2	−2.1							
[IQR]	[−4.50–−1.50]	[−4.60–−1.50]	[−4.50–−3.00]	[−3.00–−1.50]	[−3.60–−1.40]							
Body Mass Index (BMI)-for-age Z-score at the start of HPN						0.55	0.78	0.25	0.36	0.4	0.43	0.89
Median	−2.2	−2.3	−2.6	−1.9	−1.5							
[IQR]	[−3.70–−1.30]	[−3.70–−1.40]	[−3.90–−1.30]	[−3.10–0.10]	[−3.00–−1.10]							

BMI: body mass index. BW: body weight. CE: congenital enteropathy. GA: gestational age. HPN: home parenteral nutrition. IQR: interquartile range. SBS: short bowel syndrome. SMD: severe motility disorder.

## Data Availability

The research data presented in this study are available upon request from the corresponding author.

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
