# Peer review of "Epidemiological Characteristics of Pediatric Patients with Intestinal Failure in Spain: Data from the REPAFI Registry"

_nutrients, 2025, doi:10.3390/nu17233768_

Round 1

Reviewer 1 Report

Comments and Suggestions for Authors

As the authors mention, home parenteral nutrition (HPN) for prediatric patients provides advantages over remaining in the hospital or other care facility.  Of interest is understanding the peds patients who are subject for HPN.  This contribution provides data for Spain that can be compared with findings for peds HPN populations in other countries.  The following comments are provided to assist the authors in addressing what might enhance their contribution.

The registry that was established included only consented patients.  Can the authors provide information about the total number of HPN patients in Spain?  Although patient characteristics can't be included, it would be interesting to know the total numbers and the percentage represented by the present patient population.  Is this possible?

The geographical description is biased because of La Paz dominance.  

Page 6:  spell out <18 to less than 18 years of age.

Table 5 is hard to discern as presently formatted.  Other tables need attention.

Growth compared to before HPN and with normal subjects – how effective is HPN?

NEC referred to as cause of SBS in Discussion lines 36-37 and lines 50-51.  Delete redundancy.

What was nutritional status at the start of HPN?  Apparently, it was TPN.  

Curious:  Is there  way to detrmine if growth was different with HPN compared with when patients were in the hospital?  This may be outside the scope of the study.

Did any of the patients consume any oral fluids or nutrition?

Author Response

Thank you for your comments and observations, which the authors greatly appreciate, as we are convinced that they will help us improve the quality of our manuscript. Below, we address each of your points one by one.

Comment 1: The registry that was established included only consented patients.  Can the authors provide information about the total number of HPN patients in Spain?  Although patient characteristics can't be included, it would be interesting to know the total numbers and the percentage represented by the present patient population.  Is this possible?

Response 1: Unfortunately, we do not have access to this information. As we mentioned in the manuscript, there is no an official registry in Spain that includes patients receiving HPN. Therefore, we do not know the exact number of patients, nor do we have a way to determine it. This is precisely why we consider our study to be important, as we believe that the data presented are the closest possible approximation to the current reality. We are aware that not all patients who initiated HPN during this period are included, since participation in the registry is voluntary and requires patient consent. However, we believe that these data are highly representative, as all the main hospitals that care for pediatric patients with intestinal failure are included. Moreover, because the registry is anonymous, it is very likely that almost all patients or their families agreed to be included.

Comment 2: The geographical description is biased because of La Paz dominance.  

Response 2: Thank you for pointing this out. Indeed, we agree with your comment, and that is why we have addressed this point in the Discussion section. In fact, this is one of the issues identified by our society’s Intestinal Failure Working Group, as it reflects the current situation in our country due to the territorial organization of the healthcare system. La Paz Hospital is the only intestinal transplant center in Spain, and we consider it reasonable that there is a single national center performing intestinal transplants, given the very low number of such procedures carried out each year.

However, there are no officially accredited reference units for pediatric intestinal failure, as this figure does not currently exist within the system. As a result, many patients from smaller hospitals across Spain are referred to La Paz, even if they are not potential transplant candidates, simply because, from an administrative standpoint, they cannot be referred to intermediate centers that would be fully capable of managing their care. This explains why the number of patients treated at La Paz Hospital is significantly higher compared with other centers.

Comment 3: Page 6:  spell out <18 to less than 18 years of age.

Response 3: changed.

Comment 4: Table 5 is hard to discern as presently formatted.  Other tables need attention.

Response 4: Thank you for pointing this out. We have used the journal's Author services to improve the quality of both the tables and the text.

Comment 5: Growth compared to before HPN and with normal subjects – how effective is HPN?

Response 5: We are not entirely sure we understand the question. The nutritional status (weight, height, and BMI) of the subjects included in the study at the start of HPN is presented in Table 5. This table also includes a comparison with healthy subjects of the same age, using standard deviations based on WHO growth charts. We do not have anthropometric data from patients before the initiation of HPN, and regarding the growth evolution of the patients after starting HPN, those data have not yet been analyzed. Our intention is to report these findings in a future paper.

Comment 6: NEC referred to as cause of SBS in Discussion lines 36-37 and lines 50-51.  Delete redundancy.

Response 6: You are right, thank you for your observation. We have deleted the second sentence.

Comment 7: What was nutritional status at the start of HPN?  Apparently, it was TPN.  

Response 7: As I mentioned in the response 5, the nutritional status (weight, height, and BMI) of the subjects included in the study at the start of HPN is presented in Table 5.

Comment 8: Curious:  Is there  way to detrmine if growth was different with HPN compared with when patients were in the hospital?  This may be outside the scope of the study.

Response 8: Unfortunately, no. We only collected anthropometric data at birth, at the time of initiating HPN, and periodically at certain intervals after HPN was started. However, those last follow-up data have not yet been compiled.

Comment 9: Did any of the patients consume any oral fluids or nutrition?

Response 9: Yes, most patients received, in addition to parenteral nutrition, either oral or enteral nutrition. However, we did not record the percentage or the composition of the oral nutrition.

Reviewer 2 Report

Comments and Suggestions for Authors

This is an interesting and important epidemiologic study. Nevertheless, there are a few comments for improvement to be made:

Major:

1. Introduction, para 1, l. 2: Define minimum length needed in units.

2. Para 2, l. 1: what is meant  by >The low prevalence of this condition...< ?

3. Para 3, l. 6: define total number of centers, and their fraction in total IFD patient care.

4. Figure 1: translate figure to English language. Font size must be increased for readability.

5. Table 4+5: Improve design of tables, e.g. by broadening left column. It reads hard and is confusing. That data are provided as medians and IQR should be explained in the legend rather than in the table itself. In the left column the parameter and unit is fully sufficient.

6. Discussion, l. 7f: provide time points/years in Spain and other countries in comparison.

7. l. 32ff: The authors should include preterm infants, particularly those <28wk GA, who's survival rate has increased, but frequently have SBS due to necrotizing enterocolitis.

Minor:

1. Abstract, L. 14: specify 32.5 as most prevalent, as it is less than 50% of the SBS fraction.

2. L. 16+17: specify 62.7% and 23%. Is it of all or of SBS patients?

3. L. 18: indicate full range, please.

4. Results, para 1, l. 2: 10 of how many hospitals?

Discussion:

5. L. 61: change to ... treat patients with this profile.

6. L. 72: define that proportion as a %-value.

7. Table 1: was the distribution of causes equally distributed among centers? Please briefly mention in text.

8. Fig. 2: include numbers and %-values in figure.

Author Response

Thank you for your comments and observations, which the authors greatly appreciate, as we are convinced that they will help us improve the quality of our manuscript. Below, we address each of your points one by one.

Major:

Comment 1. Introduction, para 1, l. 2: Define minimum length needed in units.

Response 1. Excuse me, I’m not sure I fully understand the comment. Are you referring to the definition of intestinal failure or to that of short bowel syndrome? In the case of intestinal failure, it is defined as a critical reduction in the mass of functioning intestine, but the definition does not include length units, since there may be patients with intestinal failure who retain their entire intestine, and the failure is due to poor intestinal function, as seen in motility disorders or congenital enteropathies. The American Society for Parenteral and Enteral Nutrition (ASPEN), for example, defines pediatric intestinal failure (IF) as “the reduction of functional intestinal mass below that which can sustain life, resulting in dependence on supplemental parenteral support for a minimum of 60 days within a 74 consecutive day interval.”

In the case of short bowel syndrome, which is the most common cause of intestinal failure, there is indeed a classification based on the length of the remaining intestine, which we could include in the introduction. However, I’m not sure if that is what the comment is referring to. The most widely accepted current definition of pediatric intestinal failure is provided in Reference 1.

Comment 2. Para 2, l. 1: what is meant  by >The low prevalence of this condition...< ?

Response 2. This means that pediatric intestinal failure is a rare disease with very low prevalence, as we elaborate later in the introduction. According to reported data, prevalence ranges from 9.5 to 27 cases per million in countries similar to ours.

Comment 3. Para 3, l. 6: define total number of centers, and their fraction in total IFD patient care.

Response 3. According to the latest report from the 2020 Statistics on Specialized Healthcare Centers, there are 249 public hospitals in Spain with pediatric services, and 196 private ones. In total, 445 hospitals offer pediatric care. However, we do not know how many of these hospitals treat patients with intestinal failure. We believe that most of the centers providing such care are those participating in the study, as they include the main hospitals in the country with pediatric gastroenterology or nutrition units.

Comment 4. Figure 1: translate figure to English language. Font size must be increased for readability.

Response 4. Thank you for pointing this out. We have used the journal's Author services to improve the quality of both the tables and the text.

Comment 5. Table 4+5: Improve design of tables, e.g. by broadening left column. It reads hard and is confusing. That data are provided as medians and IQR should be explained in the legend rather than in the table itself. In the left column the parameter and unit is fully sufficient.

Response 5. As I mentioned in the previous comment, we have used the journal’s Author services in order to improve the quality of tables and we believe this has significantly improved its readability and comprehension.

Comment 6. Discussion, l. 7f: provide time points/years in Spain and other countries in comparison.

Response 6. We have added this sentence to the discussion: “Denmark was the first European country to implement home parenteral nutrition (HPN) programs, initiating them in 1970. The first case of HPN in France was reported in the early 1970s, while in the United Kingdom, initial cases emerged in the late 1970s, followed by a steady increase in subsequent decades. In other European countries, such as Belgium and Germany, the expansion of HPN occurred primarily during the 1980s and 1990s” with a new reference (21) in order to compare Spain with other european countries.

Comment 7. l. 32ff: The authors should include preterm infants, particularly those <28wk GA, who's survival rate has increased, but frequently have SBS due to necrotizing enterocolitis.

Response 7. The study includes preterm patients without any gestational age restrictions, including those with a GA < 28 weeks. Consequently, the GA of patients with short bowel syndrome was lower than that of patients with other causes of short bowel, since necrotizing enterocolitis—the most common cause of short bowel—is predominantly observed in premature infants.

Minor:

Comment 1. Abstract, L. 14: specify 32.5 as most prevalent, as it is less than 50% of the SBS fraction.

Response 1. We identified necrotizing enterocolitis as the most frequent cause of SBS. It accounted for 32.5% of all SBS cases.

Comment 2. L. 16+17: specify 62.7% and 23%. Is it of all or of SBS patients?

Response 2. It’s of SBS patients. We have clarified it in the text.

Comment 3. L. 18: indicate full range, please.

Response 3. We have added that information.

Comment 4. Results, para 1, l. 2: 10 of how many hospitals?

Response 4. As I mentioned in a previous response, in 2020 there were in Spain 445 hospitals with a pediatric unit.

Discussion:

Comment 5. L. 61: change to ... treat patients with this profile.

Response 5. Changed.

Comment 6. L. 72: define that proportion as a %-value.

Response 6. We added the percentage.

Comment 7. Table 1: was the distribution of causes equally distributed among centers? Please briefly mention in text.

Response 7. Yes, the distribution of the causes of intestinal failure (IF) was similar across all centers, with minor variations. We have mentioned this in the text.

Comment 8. Fig. 2: include numbers and %-values in figure.

Response 8. We have added them.

Round 2

Reviewer 2 Report

Comments and Suggestions for Authors

Fine by now, with one improvement lef. The revision is fine. Ther is one problem remaining, i. e. the colors of fig. 2: Don't use blue font for numbers and use lighter blue for necrotizing enterocolitis. Otherwise, it cannot be deciphered.

Author Response

Comment 1: Fine by now, with one improvement lef. The revision is fine. Ther is one problem remaining, i. e. the colors of fig. 2: Don't use blue font for numbers and use lighter blue for necrotizing enterocolitis. Otherwise, it cannot be deciphered.

Response 1: Thank you for pointing this out. We have modified the figure you mentioned in order to improve it and make it easier to understand. We hope it can now be read without any problem.